# JUTAR: Joint User-Association, Task-Partition, and Resource-Allocation Algorithm for MEC Networks

**DOI:** 10.3390/s23031601

**Published:** 2023-02-01

**Authors:** Ling Kang, Yi Wang, Yanjun Hu, Fang Jiang, Na Bai, Yu Deng

**Affiliations:** 1Information Materials and Intelligent Sensing Laboratory of Anhui Province, Anhui University, Hefei 230601, China; 2National Engineering Research Center for Agro-Ecological Big Data Analysis & Application, Anhui University, Hefei 230601, China

**Keywords:** mobile edge computing (MEC), computation offloading, joint optimization, system overhead

## Abstract

Mobile edge computing (MEC) is a promising technique to support the emerging delay-sensitive and compute-intensive applications for user equipment (UE) by means of computation offloading. However, designing a computation offloading algorithm for the MEC network to meet the restrictive requirements towards system latency and energy consumption remains challenging. In this paper, we propose a joint user-association, task-partition, and resource-allocation (JUTAR) algorithm to solve the computation offloading problem. In particular, we first build an optimization function for the computation offloading problem. Then, we utilize the user association and smooth approximation to simplify the objective function. Finally, we employ the particle swarm algorithm (PSA) to find the optimal solution. The proposed JUTAR algorithm achieves a better system performance compared with the state-of-the-art (SOA) computation offloading algorithm due to the joint optimization of the user association, task partition, and resource allocation for computation offloading. Numerical results show that, compared with the SOA algorithm, the proposed JUTAR achieves about 21% system performance gain in the MEC network with 100 pieces of UE.

## 1. Introduction

Various mobile applications facilitate people’s lives but bring very high requirements for mobile user equipment (UE). Admittedly, UE enjoys more and more powerful computing process units (CPU) nowadays, but they still can not handle the numerous computation tasks given a strict low-latency restriction due to their inherent hardware resource limits. Since the base station (BS) holds hundreds of times the computation capability of the UE, it can help UE solve heavy computation tasks. In particular, UE can offload all or part of the computation-intensive tasks to the BS, relieving its local computation burden. Based on this idea, several techniques have recently been proposed to solve the computation overload problem with respect to UE [1]. The mobile cloud computing (MCC) [2] technique works by uploading the tasks of UE to the cloud servers at the macro BS (MBS), which possesses very powerful computation capabilities and can finish the uploaded tasks within a short time. However, the transmission latency for the UE may be unacceptable if the UE suffers from a far distance from the MBS. To address this bottleneck, a mobile edge computing (MEC) [3,4] technique is further proposed as a promising approach for this issue. By deploying the small BS (SBS) involving servers at the edge of the network, each UE can upload its computation tasks to the nearest SBS, thus not only mitigating its computation load but also reducing the transmission latency.

### 1.1. Related Works

Drawing insights from MEC literature, the computation offloading strategy ranks as a dominant issue in the MEC network. Refs. [5,6,7] considered the computation offloading issue with the binary offloading strategy, in which all computation tasks for each UE are either computed locally or uploaded to servers. The binary offloading strategy enjoys low computational complexity but causes a high awaiting latency for each UE due to its principle that the computation tasks can not be processed until all of the tasks are uploaded. To overcome this drawback, the partial computation offloading strategy was proposed in [8,9,10,11], in which some computation tasks for each UE are computed locally, and other computation tasks are uploaded to servers. The partial computation offloading strategy allows performing the computation offloading and task processing in parallel, thus reducing the system latency. In particular, Refs. [8,9,10,11] merely aimed at optimizing a single key parameter indicator (KPI), e.g., energy consumption or system latency, whereas [11] investigated the joint optimization of several KPIs to achieve the best system overhead. However, Refs. [8,9,10,11] only considered MEC networks with one edge server, which bottlenecks the system’s computational capability. Fortunately, this problem can be solved by deploying densely distributed servers [12,13] in MEC networks. The main issues considered in the MEC network with densely distributed servers are user association, task partition, and resource allocation [14,15,16,17]. In particular, Ref. [14] solved the user association problem according to the bandwidth, power, and interference. Ref. [15] considered the user association issue according to the data size of the uploaded tasks. Refs. [16,17] jointly considered the task partition and resource allocation issues to not only balance the overload of each server but also exploit the computational capability of each server more efficiently. Recently, deep learning techniques were extensively employed to optimize the computation offloading problem [18,19,20,21,22]. Ref. [18] used the genetic algorithm (GA) to decide whether the tasks of each UE are offloaded to SBS or MBS. Ref. [19] enhanced the decision strategy by comprehensively considering the GA and each UE’s offloading prior probability. Ref. [20] relied on deep reinforcement learning (DRL) to solve the resource allocation issue. Ref. [21] employed the GA to save energy consumption and further considered each UE’s mobility. Ref. [22] exploited the LSTM network to further improve the system performance by considering each UE’s direction.

### 1.2. Motivation and Contributions

Thus far, the existing computation offloading schemes still suffer from several concerns. First, the user association should not only consider the bandwidth, power, and interference for each UE but also evaluate the data size of the tasks and the channel quality [14,15]. Second, the traditional algorithms employed the task partition and resource allocation shows low convergence speeds, which bottleneck the system performance [16,17]. Finally, the existing computation offloading schemes usually optimize the user association, the task partition and the resource allocation problem, respectively, which could miss the optimal solution for the overall MEC network [9,12]. The performance of the computation offloading scheme is dominated by the user-association together with the task-partition and the resource-allocation strategy, and this knowledge motivates us to think of an approach to jointly optimize the user-association, task-partition, and resource-allocation issues. In this paper, we propose a joint user-association, task-partition, and resource-allocation (JUTAR) algorithm to solve the computation offloading problem in the MEC network with densely distributed servers. The main contributions of this work are listed as follows:We build up the optimization function with respect to the joint user-association, task-partition, and resource-allocation issues given an MEC network with massive servers. With the joint optimization of these problems, the optimization function could explore better results for the realistic MEC network.We define a user-association metric, which comprehensively considers the distance and overload of each UE and the target SBS, to indicate the user-association for each UE. In addition, we employed the smooth approximation to further simplify the optimization function.We propose using the particle swarm algorithm (PSA) to find the optimal results of the optimization function. The PSA can heuristically find the optimal solutions for the function and contribute to better system performance.

Numerical results also demonstrate that the proposed JUTAR algorithm enjoys the lowest system overhead compared with the existing computation offloading schemes given different parameters (e.g., number of users, data size of tasks).

The rest of this paper is organized as follows. Section 2 introduces the system model and the problem formulation of MEC networks. Section 3 discusses the details of the proposed JUTAR algorithm. Section 4 provides numerical results to validate the proposed algorithm. Section 5 concludes this work.

## 2. System Model and Problem Formulation

Figure 1 shows the model of the MEC network, involving an MBS, several SBSs, and various types of UE. Each BS is equipped with an MEC server. The SBSs and UE are randomly distributed within the coverage of the MBS. In addition, all of the SBSs are connected to the MBS via wired links. Supposing there are M+1 BSs (including the MBS), and *N* pieces of UE, let $S=\left\{S_{0},S_{1},\cdots ,S_{M}\right\}$ and $U=\left\{u_{1},u_{2},\cdots ,u_{N}\right\}$ denote the set of the BSs and the UE, respectively, where $S_{0}$ represents the MBS, $S_{m}$ (m∈{1,2,...,M}) denotes the *m*-th SBS, and $u_{n}$ (n∈{1,2,...,N}) denotes the *n*-th UE. Although each user could detect several SBSs in this scenario, we only consider the case that each user associates with one SBS for simplicity. $a_{m,n}$ indicates that UE *n* is associated with SBS *m*, and we have
(1)$$a_{m,n}=\left\{\begin{array}{c} 1,\;\mathrm{if}\;n\text{-}\mathrm{th}\;\mathrm{UE}\in m\text{-}\mathrm{th}\;\mathrm{SBS},\\ 0,\;\mathrm{otherwise}. \end{array}\right.$$

In the MEC network system, a computational task can be divided into several independent subtasks, which are addressed by the MEC servers and the local devices, respectively. The task requirement of the *n*-th UE is denoted as $I_{n}=<d_{n},c_{n},t_{n}^{\max}>$, where $d_{n}$ is the data size of the task, $c_{n}$ is the CPU clock cycles required to process a bit of data in the task, $t_{n}^{\max}$ is the maximum latency of this task. Therefore, the task $I_{n}$ requires $c_{n}d_{n}$ clock cycles for the *n*-th UE.

Generally, computational offloading has two different solutions. On the one hand, as Figure 2a shows, if the UE is directly connected to the MBS, its computational task is divided into two parts: (1) local computational task, denoted as $x_{n}^{l}$, and (2) remote computational task, denoted as $x_{n}^{c}$, where $x_{n}^{l}+x_{n}^{c}=1$. On the other hand, if the UE is connected into an SBS, its computational tasks are processed in part by the local device (local computation, denoted as $x_{n}^{l}$), in part by the SBS (computational offloading, denoted as $x_{m,n}^{e}$), and in part by the MBS (further computational offloading, denoted as $x_{n}^{c}$). Figure 2b shows the offloading strategy of the UE corresponding to an SBS, and the processed tasks satisfy $x_{n}^{l}+x_{m,n}^{e}+x_{n}^{c}=1$. Since the local tasks and the offloading tasks are processed in parallel, the system latency is dominated by the part with the highest latency. However, energy consumption is the accumulation of energy consumption for different parts.

### 2.1. Local Computing Model

Let $t_{n}^{l}$ denote the latency of the local tasks, which can be written as
(2)$$t_{n}^{l}=\frac{x_{n}^{l}c_{n}d_{n}}{f_{n}^{l}},$$
where $f_{n}^{l}$ is the computational capability of the *n*-th UE, measured in CPU clock cycles per second. Let ς denote the energy coefficient depending on CPU chip architecture, then the power consumption of the CPU is expressed as $\varsigma {\left(f_{n}^{l}\right)}^{3}$. Hence, the execution energy for this task (denoted as $E_{n}^{l}$).
(3)$$E_{n}^{l}=x_{n}^{l}c_{n}d_{n}\varsigma {\left(f_{n}^{l}\right)}^{2}.$$

### 2.2. Computing Model in SBS

The tasks with respect to the *n*-th UE and processed in the *m*-th SBS are divided into two parts. In particular, the first part will be offloaded into a corresponding MEC server, and the second part is processed by the SBS itself. Let the notation *B*, $\sigma^{2}$, $p_{n}$ and $h_{m,n}$ denote the system bandwidth, noise power, transmit power of the *n*-th UE, and channel coefficient between the *n*-th UE and *m*-th SBS, respectively, then the transmission rate of the *n*-th UE is
(4)$$r_{m,n}=B\log_{2}\left(1+\frac{P_{n}h_{m,n}}{\sigma^{2}}\right)\;\left(\mathrm{bits}/s\right).$$

In addition, let the notation $f_{m,n}$ represent the computational capacity (clock cycles per second) of the *m*-th SBS, and the maximum computational capacity of the *m*-th SBS is $f_{m,n}^{\max}$. With the notation defined above, the latency $t_{m,n}^{e}$ and energy consumption $E_{m,n}^{e}$ of the tasks with respect to the *n*-th UE and processed in the *m*-th SBS can be written as
(5)$$t_{m,n}^{e}=a_{m,n}x_{n}^{e}\left(\frac{d_{n}}{r_{m,n}}+\frac{c_{n}d_{n}}{f_{m,n}}\right),$$
(6)$$E_{m,n}^{e}=a_{m,n}x_{n}^{e}\left(p_{n}\frac{d_{n}}{r_{m,n}}+c_{n}d_{n}e_{s}\right),$$
where $e_{s}$ is the energy consumption of *m*-th SBS per CPU cycle clock. It is noted that the latency and energy consumption of the feedback operation from the MEC sever are neglected since the data size of the feedback is negligible compared with the transmitted data [7].

### 2.3. Computing Model for the MBS

The latency of the tasks associated with $x_{n}^{c}$ consists of the transmission latency and the processing latency. In detail, if the *n*-th UE is directly connected to the MBS, the transmission rate is
(7)$$r_{0,n}=B\log_{2}\left(1+\frac{P_{n}h_{0,n}}{\sigma^{2}}\right)\;\left(\mathrm{bits}/s\right),$$
where $h_{0,n}$ is the channel gain between the *n*-th UE and the MBS. On the contrary, if the *n*-th UE is linked with an SBS, the transmission rate is $r_{m,n}$ defined above. In addition to the latency transmitting the data from the *n*-th UE to the *m*-th SBS, the overall transmission latency also involves the latency offloading the data from the *m*-th SBS to the MBS, which is represented as $\gamma x_{n}^{c}a_{m,n}d_{n}$. Here, γ is a factor of the transmission via the wired line. Let the notation κ be the indicator about whether the *n*-th UE is directly connected to the MBS (1 for YES and 0 for NO), then the overall latency about the tasks $x_{n}^{c}$ is
(8)$$t_{n}^{c}=\frac{x_{n}^{c}d_{n}}{r_{n}}+\left(1-\kappa \right)\gamma x_{n}^{c}a_{m,n}d_{n}+\frac{x_{n}^{c}c_{n}d_{n}}{F_{c}},$$
where $F_{c}\;\left[\mathrm{clock}\;\mathrm{cycles}/s\right]$ is the computing capacity of the MEC server corresponding to the MBS [10]. Following the same principle as the latency analysis, the overall energy consumption of the tasks associated with $x_{n}^{c}$ is
(9)$$E_{n}^{c}=P_{n}\frac{x_{n}^{c}d_{n}}{r_{n}}+\left(1-\kappa \right)\beta \gamma x_{n}^{c}a_{m,n}d_{n}+x_{n}^{c}c_{n}d_{n}e_{c},$$
where β is the transmission power consumption via a wired line $e_{c}$ is the energy consumption per CPU clock cycle of the MBS [16].

Thus far, the overall latency of the tasks for the *n*-th UE is
(10)$$t_{n}=\max\left(t_{n}^{l},t_{m,n}^{e},t_{n}^{c}\right),$$
and the overall energy consumption for processing the tasks of the *n*-th UE is expressed as
(11)$$E_{n}=\left(E_{n}^{l}+E_{m,n}^{e}+E_{n}^{c}\right).$$
Table 1 concludes the notations used in this paper.

## 3. Methodology

Since the latency and the energy consumption serve as the key parameter indicator (KPI) for the MEC network, designing an offloading strategy that can balance those two KPIs becomes demanding. Considering the trade-off of the latency and the energy consumption, the optimization problem of the computation offloading is formulated as
(12)$$\begin{array}{ccc} P1: & \min_{a_{m,n},x_{n},f_{m,n}}\sum_{n=1}^{N}\left(t_{n}+\lambda E_{n}\right),\\ \; & s.t.\;t_{n}\leq t_{n}^{\max}, & \;\left(C1\right)\;\\ \; & \sum_{m=1}^{M}a_{m,n}\leq 1,a_{m,n}\in \left\{0,1\right\}, & \;\left(C2\right)\;\\ \; & x_{n}^{l}+x_{m,n}^{e}+x_{n}^{c}=1,\forall m\in M,n\in N, & \;\left(C3\right)\;\\ \; & f_{m,n}>0,\forall m\in M,n\in N, & \;\left(C4\right)\;\\ \; & \sum_{n\in N}f_{m,n}\leq f_{m}^{\max},\forall m\in M,n\in N & \;\left(C5\right)\; \end{array}$$

Here, λ denotes the weight coefficients corresponding to the latency and energy consumption, and (C1) limits the maximum latency. (C2) guarantees that each UE can be associated with at most one SBS. (C3) promises that the overall tasks of the *n*-th UE are normalized as one. (C4) and (C5) circumvent the scenarios in that the required computational resource of the connected UE exceeds the overall computational capacity for the *n*-th SBS. Unfortunately, the optimal solution of the P1 problem can not be computed directly since there are several strongly coupled variables, and the problem is built by a non-convex nonlinear function. To this end, heuristic methods are employed in this section. In particular, the P1 problem is first simplified by associating each UE with the appropriate SBS and then approximated by utilizing a smooth approximation equation. Finally, a PSA algorithm is developed to find the optimal solution to the problem.

### 3.1. User Association

Supposing the network involves *M* SBSs and *N* users, let A denote the user-association matrix with dimensions of M×N, written as
(13)$$\left[\begin{array}{cccc} a_{1,1} & a_{1,2} & \cdots & a_{1,N}\\ a_{2,1} & a_{2,2} & \cdots & a_{2,N}\\ \vdots & \vdots & \vdots & \vdots \\ a_{M,1} & a_{M,2} & \cdots & a_{M,N} \end{array}\right]$$
where $a_{i,j}$ represents whether the *j*-th user is associated with the *i*-th SBS. Since the *i*-th user may be covered by several SBSs at the same time, the *i*-th column of the A could involve *K*, 1≤K≤M unitary elements. Let $\Phi_{n}=\left\{\phi_{1},\phi_{2},...,\phi_{K}\right\}$ denote a vector consisting of the index of the unitary element in the *n*-th column of the matrix A. The user association for each UE should comprehensively consider: (1) the distance between the UE and the target SBS, and (2) the overload of the SBS. Based on this, we denote $b_{\phi_{k},n}$ as the association metric with respect to each SBS. The $b_{\phi_{k},n}$ is illustrated as
(14)$$b_{\phi_{k},n}=\frac{h_{\phi_{k},n}}{1+\frac{{\mathrm{load}}_{\phi_{k}}}{\sum_{k}{\mathrm{load}}_{\phi_{k}}}},k=\left\{1,2,\cdots ,K\right\}.$$
where ${\mathrm{load}}_{\phi_{k}}$ denotes the overload of the $\phi_{k}$-th SBS and $h_{\phi_{k},n}$ is the channel coefficient between the *n*-th UE and the $\phi_{k}$-th SBS. The channel coefficient $h_{\phi_{k},n}$ follows the Rayleigh fading channel model, and is computed by
(15)$$h_{\phi_{k},n}=140.7+36.7\;\log d,$$
where *d* is the distance between each pair of UE and the target SBS. It is observed that the larger the $b_{\phi_{k},n}$, the more reliable the SBS. Hence, each UE will be associated with the corresponding SBS with the largest association metric. Since the UE may also be covered by the MBS, they can be categorized into two groups. Let $N_{m,0}$ denote the first group of UE, in which the offloading tasks for each UE can only be processed by the MBS. Denote $N_{m,0}^{c}$ as the second group of UE, which is the complementary set of $N_{m,0}$. The offloading tasks for each UE in $N_{m,0}^{c}$ are addressed in part by the SBS and in part by the MBS. Hence, the P1 optimization problem can be simplified as
(16)$$\begin{array}{cc} P2: & \min_{x_{n}}\sum_{n\in N_{m,0}}\left(\max\left(t_{n}^{l},t_{n}^{c}\right)+\lambda \left(E_{n}^{l}+E_{n}^{c}\right)\right)+\\ \; & \min_{x_{n},f_{m,n}}\sum_{n\in N_{m,0}^{c}}\left(\max\left(t_{n}^{l},t_{m,n}^{e},t_{n}^{c}\right)+\lambda E_{n}\right),\\ \; & s.t.\;\;\left(C1\right)\;-\;\left(C5\right)\;. \end{array}$$

For simplicity, we use $O_{M}$ to denote the optimization function corresponding to the computation offloading tasks processed by the MBS and $O_{S}$ to denote the optimization function corresponding to the computation offloading tasks addressed by the SBS. Therefore, we have
(17)$$\left\{\begin{array}{c} O_{M}=\min_{x_{n}}\sum_{n\in N_{m,0}}\left(\max\left(t_{n}^{l},t_{n}^{c}\right)+\lambda \left(E_{n}^{l}+E_{n}^{c}\right)\right),\\ O_{S}=\min_{x_{n},f_{m,n}}\sum_{n\in N_{m,0}^{c}}\left(\max\left(t_{n}^{l},t_{m,n}^{e},t_{n}^{c}\right)+\lambda E_{n}\right). \end{array}\right.$$

### 3.2. Smooth Approximation

Although the function $O_{M}$ can be solved by employing the linear optimization toolbox, the solution of the P2 problem is still unavailable due to the nonlinear optimization function $O_{S}$ involving the variables $x_{n}$ and $f_{m,n}$. Hence, heuristic approaches (e.g., PSA algorithm) can be employed to find optimal the solutions to $O_{S}$. However, the max function in $O_{S}$ involves tree variables, which makes it hard to derive the optimal results theoretically, hindering the application of the heuristic approaches. Fortunately, the max function can be eliminated by the smooth approximation function for the mathematical uniform norm [23]. Let $O_{S}^{\prime }=\min_{x_{n},f_{m,n}}\sum_{n\in N_{m,0}^{c}}\max\left(t_{n}^{l},t_{m,n}^{e},t_{n}^{c}\right)$, and then it can be rewritten as
(18)$$O_{S}^{\prime }=\min_{u\in \Phi^{n}}\max\left(u\right)=\min_{u\in \Phi^{n}}{∥u∥}_{\infty },$$
where u is the vector involving all of the possible choices with respect to $\left(t_{n}^{l},t_{m,n}^{e},t_{n}^{c}\right)$. According to the property of the uniform norm, by introducing the entropy function, Equation (Equation 18) can be approximated by
(19)$$\min_{u\in \Phi^{n}}F_{\mu }\left(u\right)=\mu \ln\left[\left(\sum_{i=1}^{n}\left(e^{\frac{u_{i}}{\mu }}+e^{-\frac{u_{i}}{\mu }}\right)\right],\right.$$
where μ is the smoothing factor. Therefore, the $\max\left(t_{n}^{1},t_{m,n}^{e},t_{n}^{c}\right)$ can be rewritten as
(20)$$\mu \ln\left[e^{\frac{t_{n}^{1}}{\mu }}+e^{-\frac{t_{n}^{1}}{\mu }}+e^{\frac{t_{mn}^{e}}{\mu }}+e^{-\frac{t_{mn}^{e}}{\mu }}+e^{\frac{t_{n}^{c}}{\mu }}+e^{-\frac{t_{n}^{c}}{\mu }}\right].$$

Replacing the $\max\left(t_{n}^{1},t_{m,n}^{e},t_{n}^{c}\right)$ with (Equation 20), the P2 problem can be converted into P3 problem, illustrated as
(21)$$\begin{array}{cc} P3: & \min_{x_{n}}\sum_{n\in N_{m,0}}\left(\max\left(t_{n}^{l},t_{n}^{c}\right)+\lambda \left(E_{n}^{l}+E_{n}^{c}\right)\right)+\\ \; & \min_{x_{n},f_{m,n}}\sum_{n\in N_{m,0}^{c}}\left(\mu \ln\left[e^{\frac{t_{n}^{l}}{\mu }}+e^{-\frac{t_{n}^{l}}{\mu }}+e^{\frac{t_{m,n}^{e}}{\mu }}+\right.\right.\\ \; & \;\left.\left.e^{-\frac{t_{m,n}^{e}}{\mu }}+e^{\frac{t_{n}^{c}}{\mu }}+e^{-\frac{t_{n}^{c}}{\mu }}\right]+\lambda E_{n}\right)\\ \; & s.t.\;\;\left(C1\right)\;-\;\left(C5\right)\;. \end{array}$$

### 3.3. Optimized with PSA Algorithm

With the derivations above, the joint optimization problem for the computation offloading in a single SBS scenario is simplified as the P3 function, which can be solved with the PSA algorithm [24]. The PSA algorithm enjoys the advantages of rapid convergence and low complexity. In addition, it can circumvent the local optimal results and achieve the approximated global optimal point. The details of employing the PSA algorithm to solve the P3 problem is listed as follows:

(1) Initialization: Let $\Theta_{m}=\left\{\theta_{1},\theta_{2},...,\theta_{|N_{m,0}^{c}|}\right\}$ denote a particle for the *m*-th single SBS, where $\theta_{k}$ represents the computation offloading strategy for the *k*-th user involved in set $N_{m,0}^{c}$. For the *k*-th user, recall that $x_{k}^{l}$ and $x_{k}^{e}$ denote the tasks processed locally and the tasks offloaded from the user to the single SBS, respectively. Given $x_{k}$ and $f_{m,k}$, $\theta_{k}$ can be denoted as $\theta_{k}=\left(x_{k}^{l},x_{k}^{e},f_{m,k}\right)$. Figure 3 shows a toy example for $\Theta_{m}$ with $|N_{m,0}^{c}|=3$. For initialization, it is assumed that there are *Q* particles. In addition, let $V_{m}=\left\{v_{1},v_{2},...,v_{|N_{m,0}^{c}|}\right\}$ represent the velocity towards to the particle $\Theta_{m}$. In particular, the velocity with respect to the *k*-th user is $v_{k}=\left(v_{k}^{l},v_{k}^{e},v_{k}^{f}\right)$, where $v_{k}^{l}$, $v_{k}^{e}$, and $v_{k}^{f}$ are initialized with random values. It is noted that all of the initial values must satisfy the constraints in the P1 problem.

(2) Update Fitness function: According to the formula of the P3 problem, the *Fitness function* is denoted as
(22)$$\begin{array}{cc} \mathrm{Fit} & =\sum_{n\in N_{m,0}^{c}}\left(\mu \ln\left[e^{\frac{t_{n}^{l}}{\mu }}+e^{-\frac{t_{n}^{l}}{\mu }}+e^{\frac{t_{m,n}^{e}}{\mu }}+e^{-\frac{t_{m,n}^{e}}{\mu }}+\right.\right.\\ \; & \;\left.\left.e^{\frac{t_{n}^{c}}{\mu }}+e^{-\frac{t_{n}^{c}}{\mu }}\right]+\lambda E_{n}\right)+\eta . \end{array}$$
where η is the penalty function for the particles. Once the particle violates the constraints, it will be penalized by the penalty function. According to the constraint in P1 problem, η is computed by
(23)$$\eta =\sum_{n=1}^{N_{m,0}^{c}}\theta_{n}\left(\max\left(0,\sum_{n\in N}f_{m,n}-f_{m}^{\max}\right)\right).$$
where $\theta_{n}$ is the penalty factor. It is emphasized that the better the offloading strategy, the smaller the fitness function.

(3) Update particle: In the *t*-th iteration, $\Theta_{m}$ and $V_{m}$ will be updated as
(24)$$\left\{\begin{array}{cc} V_{m}\left(t+1\right)= & W\cdot V_{m}\left(t\right)+C_{1}\cdot R_{1}\cdot \left(E_{\mathrm{opt}}-\Theta_{m}\left(t\right)\right)\\ \; & +C_{2}\cdot R_{2}\cdot \left(G_{\mathrm{opt}}-\Theta_{m}\left(t\right)\right),\\ \Theta_{m}\left(t+1\right)= & \Theta_{m}\left(t\right)+V_{m}\left(t+1\right), \end{array}\right.$$
where:$C_{1}$ and $C_{2}$ are two acceleration factors,*W* is a constant weight of the inertia,$R_{1}$ and $R_{2}$ are random factors chosen from [0,1],$E_{\mathrm{opt}}$ is the optimal position for the *m*-th particle in the single SBS scenario,$G_{\mathrm{opt}}$ is the optimal position for *m*-th particle in the global MEC network.

(4) Terminate algorithm: If the maximum number of the iterations is achieved, the algorithm will be terminated. On the contrary, the algorithm step (2) and step (3) are repeated.

### 3.4. Proposed JUTAR Algorithm

Thus far, the computation offloading problem in the single SBS-based scenario has been solved. However, there are multiple SBSs and massive users in the realistic MEC network. Therefore, the JUTAR algorithm is proposed in this subsection to solve the computation offloading problem in the overall MEC network.

Algorithm 1 shows the details of the JUTAR algorithm. In particular, first, each user in the network is associated with a single SBS according to the association metric computed by (Equation 14). Second, the problem corresponding to computation offloading tasks directly processed by the MBS is solved by employing the linear toolbox. Third, the computation offloading problem for each user in the single SBS scenario is transformed from the P2 problem to the P3 problem by means of smooth approximation. Finally, the results of the problem associated with the tasks offloaded to the SBS are acquired by utilizing the PSA algorithm. With the JUTAR algorithm, the P1 problem in the overall MEC network is converted into the P4 problem, written as
(25)$$\begin{array}{cc} P4: & \min_{x_{n},f_{m,n}}\sum_{m}\sum_{n\in N_{m}^{\mathrm{rec}}}\left(\max\left(t_{n}^{l},t_{m,n}^{e},t_{n}^{c}\right)+\right.\\ \; & \;\left.\lambda \left(E_{n}^{l}+E_{m,n}^{e}+E_{n}^{c}\right)\right)+\\ \; & \min_{x_{n}}\sum_{n=1}^{N_{1}}\left(\max\left(t_{n}^{l},t_{n}^{c}\right)+\lambda \left(E_{n}^{l}+E_{n}^{c}\right)\right),\\ \; & s.t.\;\;\left(C1\right)\;-\;\left(C5\right)\;. \end{array}$$

Note that although the P4 problem is derived from the P1 problem by simplifying the scenario of the network, it converges to the same results as the P1 problem [25].
**Algorithm 1:** JUTAR Algorithm
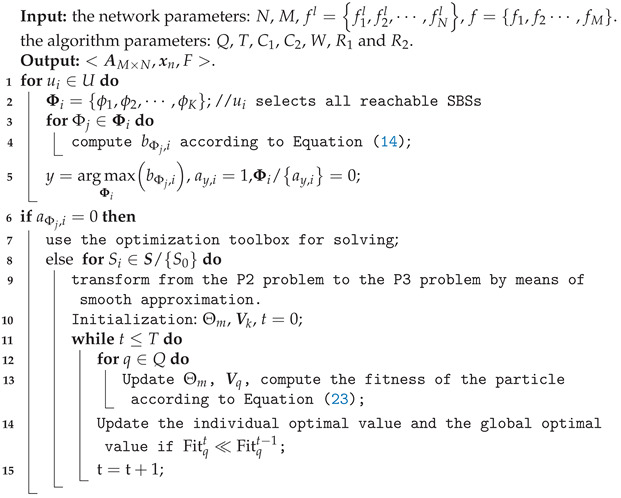



## 4. Numerical Results

In this section, we provide numerical results to demonstrate the performance of the proposed JUTAR algorithm for the MEC network. We consider a network with coverage of 1.2km×1.2km, involving one MBS, multiple SBSs, and multiple users. The MBS locates on the central point of the network, and the SBSs, together with the users, are randomly distributed. We follow the setup of the parameters in [18]. In particular, the computation input data size of subtask is $d_{n}\in \left[10,12\right]\;\mathrm{Kbits}\;$, and the number of CPU cycles to execute one bit of the task is $c_{n}\in \left[1000,1200\right]\;\mathrm{cycles}/\mathrm{bit}$. The computational capabilities of the local device is $f_{n}^{l}\in \left[4,5\right]\times 10^{8}\;\mathrm{cycles}/s$, the maximum computational capabilities of the SBS is $f_{m}^{l}\in \left[8,10\right]\times 10^{9}\;\mathrm{cycles}/s$, and the maximum computational capabilities of MBS is $F_{c}=15\times 10^{9}\;\mathrm{cycles}/s$. The transmit power for each user is set as 0.1 W. A Rayleigh fading channel is employed, and the channel coefficients are computed by (Equation 15). In addition, we set the energy coefficient with respect to the CPU as $\varsigma =10^{-28}$. The energy consumption of both the SBS and the MBS are set as $e_{s}=e_{c}=0.02$ W/GHz. Since the latency and the energy consumption use the dominant roles to measure the computation offloading algorithms, we comprehensively consider these two KPIs and propose the system overhead Ω as a new metric. The system overhead Ω is computed by
(26)$$\Omega =\sum_{n=1}^{N}\left(t_{n}+\lambda E_{n}\right).$$

Without special explanation, the parameter λ is set as λ=0.5. Consequently, the smaller the system overhead Ω, the better the computation offloading algorithm. To show the performance advantages of the proposed JUTAR algorithm, we provide comparison results of the existing algorithms as follows:All local processing (ALP) algorithm [8]: all tasks of the user are processed locally.Partial offloading strategy (POS) algorithm [8]: partial tasks of each UE are offloaded into either SBS or MBS based on a possible user association.All MBS processing (AMP) algorithm [10]: the overall tasks of each UE can only be uploaded to the MBS.GA [18]: partial tasks of each UE are uploaded to either SBS or MBS, which is decided according to the genetic algorithm.Priority offloading mechanism with joint offloading proportion and transmission (PROMOT) algorithm [19]: partial tasks of each UE are offloaded into SBS or MBS according to not only the GA algorithm but also its offloading prior probability.

Figure 4 shows the comparison results of the system overhead for the proposed JUTAR algorithm and other algorithms with different amounts of UE and date size of tasks. In particular, Figure 4a shows the system performance of the proposed JUTAR algorithm with various amounts of UE. It is observed that compared with existing algorithms, the proposed JUTAR algorithm shows comparable performance with a relatively small amount of UE but achieves the best results with a relatively large amount of UE. In particular, compared with the SOA GA algorithm, the proposed JUTAR algorithm saves about 21% system overhead given 100 pieces of UE in the MEC network. Figure 4b reports the system performance of the proposed JUTAR algorithm with various data sizes of tasks. It can be seen that the curves in Figure 4b have a consistent trend with that in Figure 4a. Compared with the GA algorithm, the proposed JUTAR algorithm saves about 33% system overhead when the offloaded tasks hold the size of 10,000 bits.

Figure 5 illustrates the system performance of the proposed JUTAR algorithm with 10–100 servers. From Figure 5, we can draw two observations. First, the system overhead of the proposed JUTAR algorithm degrades rapidly by increasing servers in the region with a relatively small number of servers but achieves a performance floor in the region with a relatively large number of servers. Second, although the proposed JUTAR algorithm suffers from a performance floor with a large number of servers, it still enjoys the best performance compared with the existing algorithms. From the numerical results reported in Figure 4 and Figure 5, we can conclude that given a different amounts of UE, servers, and data sizes of tasks, the proposed JUTAR algorithm holds the best system performance compared with the existing algorithms.

## 5. Conclusions

In this paper, we investigated the computation offloading problem in MEC networks with multiple SBSs and multiple pieces of UE. We proposed a JUTAR algorithm to solve the problem by jointly optimizing the user-association, task-partition, and resource-allocation issues. In particular, first, we simplified the network from a multiple SBS-based scenario to a single SBS-based scenario by employing the proposed user-association scheme. Second, for the single SBS-based scenario, we transformed the nonlinear optimization problem into the linear problem by means of the smooth approximation. Finally, we solved the linear problem in the single SBS-based scenario by utilizing the PSA algorithm. Given a different amount of UE, servers, and data size of tasks, our algorithm shows the smallest system overhead compared with the existing algorithms. In future work, we will optimize the complexity of the algorithm and further consider the mobility of each UE for realistic MEC scenarios.

## Figures and Tables

**Figure 1 sensors-23-01601-f001:**
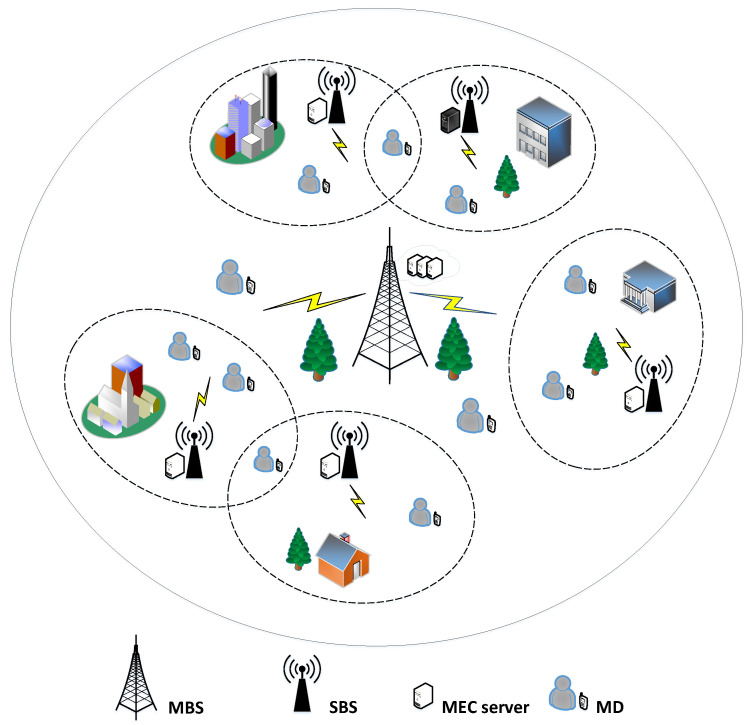
System model of the MEC network.

**Figure 2 sensors-23-01601-f002:**
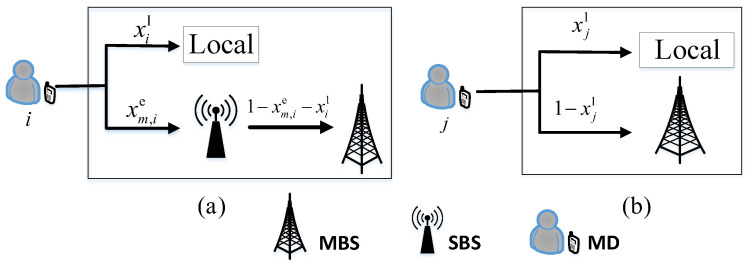
Computational offloading strategies for the UE in different scenarios: (**a**) only associated with an MBS, (**b**) associated with the SBS and MBS.

**Figure 3 sensors-23-01601-f003:**
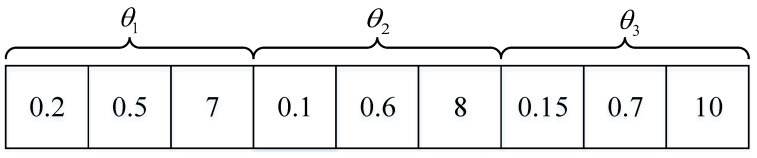
The position of the *k*-th particle.

**Figure 4 sensors-23-01601-f004:**
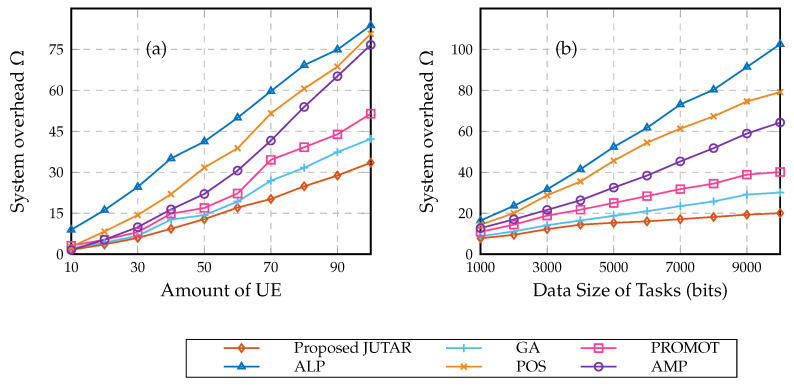
Comparison results of the system overhead vs. (**a**) amount of UE, (**b**) data size for the proposed JUTAR algorithm [8,10,18,19].

**Figure 5 sensors-23-01601-f005:**
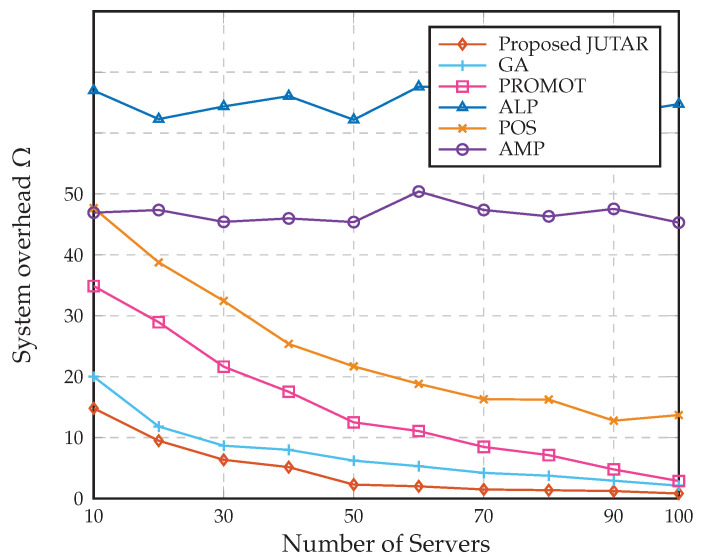
Comparison results of the system overhead vs. number of servers for the proposed JUTAR algorithm.

**Table 1 sensors-23-01601-t001:** Notations in this work.

Notations	Description
$u_{n}$	*n*-th UE
$S_{m}$	*m*-th SBS
$a_{m,n}$	Indicator of if the *n*-th UE is associated with the *m*-th SBS
$d_{n}$	Data size of the task
$c_{n}$	Required CPU clock cycles to process a bit of data in the task
$t_{n}^{\max}$	Maximum tolerance latency of this task
$t_{n}^{l}$	Latency of the local tasks
$E_{n}^{l}$	Energy consumption of the local tasks
$f_{n}^{l}$	Computational capability of the *n*-th UE
$r_{m,n}$	Transmission rate of the *n*-th UE
$t_{m,n}^{e}$	Latency of the tasks of the *n*-th UE and processed in the *m*-th SBS
$E_{m,n}^{e}$	Energy consumption of the tasks of the *n*-th UE and processed in the *m*-th SBS
$t_{n}^{c}$	Latency of the tasks of the *n*-th UE and processed in the MBS
$E_{n}^{c}$	Energy consumption of the tasks of the *n*-th UE and processed in the MBS
$t_{n}$	Overall latency of the tasks for the *n*-th UE
$E_{n}$	Overall energy consumption of the tasks of the *n*-th UE
$x_{n}$	Task offloading ratio
$b_{\phi_{k},n}$	UE association metric
${\mathrm{load}}_{\phi_{k}}$	Overload of the $\phi_{k}$-th SBS

## Data Availability

Not applicable.

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
