# Peer review of "JUTAR: Joint User-Association, Task-Partition, and Resource-Allocation Algorithm for MEC Networks"

_sensors, 2023, doi:10.3390/s23031601_

Round 1

Reviewer 1 Report

In this paper, JUTAR algorithm is proposed to solve the computation offloading problem in the MEC network with densely distributed servers. The smooth approximation of the uniform norm and the particle swarm algorithm (PSA) are utilized to find the optimal solutions. The proposed algorithm is a good extension from the existing literature, however, there are several missing details to be clarified to further strengthen the paper:

1. The difference from previous work in literature can be highlighted. The several concerns mentioned in Line 61-68 should be summarized from some given literatures. Furthermore, in section 4, the methods proposed in [8], [11], [25] and [26] are compared with JUTAR algorithm. Please also describe the difference between them in literature.  

2. Some presentation needs to be checked carefully. For example, in the sentences followed (2), the variable zeta should be in next sentence. And the word sensible in the more sensible the SBS is.  

3. In (14), should the association metric be related to the distance? There is no distance in (14).

4. Please give detailed description of the Nm,0^c. It makes some much confusion in (15) and (16). The authors denote the users whose tasks are directly processed by the MBS by Nm,0. But why there are En^c and t^c in the first part of (15). En^c and t^c contains the process from UE to SBS.

5. Please give the reason of using the smooth approximation function. Without the smooth approximation process, is it can still be solved by PSO?

6. Please give the definition of theta_n in (22). 

Reviewer 2 Report

+The proposed approach is evaluated and results are presented. But some requirements need to improve the manuscript.

+The related work is not up-to-date since the latest reference is merely up to 2019. Please add more newer references. 

+In the abstract author need to mention the contribution by % that how much your proposal is better.

+All the mathematical equation used in this paper need proper citation with clear meaning of the variable used.

+Providing free access to the code is also helpful (like GitHub).

+ I need to see in the results  (plots) the number of UEs can be  10-20-30-40-50-60-70-80-90-100, the data size of task can be till 10000 and the number of server till 100. And we can see if it work also in large scale.

Reviewer 3 Report

The topic is interesting and relevant to the field. The problem is important but can be more well motivated by corelating the research gaps in the literature. The proposed methods are well-explained. The experiments are convincing as the numerical analysis is presented. However, some major improvements are required, as suggested below.

·       Abstract needs improvements, as its not indicating the research gap and novelty of the technique.

·       In the introduction line 62, the authors said that “First, the user association should not only consider the bandwidth, power, and interference for each UE, but also evaluate the data size of the tasks and the channel quality. Second, the traditional algorithms employed for the task partition and resource allocation shows low convergence speeds, which bottleneck the system performance”. There are several ways that consider the data sizes of user tasks and channel quality. For instance, see the following studies that consider data sizes and channel health while taking offloading decisions. Pls add and cite them in the manuscript and adjust your contributions statements in the abstract/intro/literature accordingly.

o   Li, Ji, Hui Gao, Tiejun Lv, and Yueming Lu. "Deep reinforcement learning based computation offloading and resource allocation for MEC." In 2018 IEEE Wireless communications and networking conference (WCNC), pp. 1-6. IEEE, 2018.

o   Xiao, Z., Dai, X., Jiang, H., Wang, D., Chen, H., Yang, L., & Zeng, F. (2019). Vehicular task offloading via heat-aware MEC cooperation using game-theoretic method. IEEE Internet of Things Journal7(3), 2038-2052.

o   Mustafa, Ehzaz, Junaid Shuja, Ali Imran Jehangiri, Sadia Din, Faisal Rehman, Saad Mustafa, Tahir Maqsood, and Abdul Nasir Khan. "Joint wireless power transfer and task offloading in mobile edge computing: a survey." Cluster Computing 25, no. 4 (2022): 2429-2448.

·       There are more than 20 equations in the manuscript. It is suggested to keep the formulas explaining the main idea behind system modelling and move the rest of the equations to an Appendix section as the end.

·       A table of Notations and Definitions should be added that explains all symbols used in the manuscript.

·       The authors provided some insights from the MEC literature related to their problem in the introduction section. However, a separate literature review section is strongly recommended to add that must explain and compare (8-10) relevant studies in last 3 years. In the current list references the articles that have less, or no relevance should be replaced by the relevant ones. Some examples are given at the end.

·       Problem formulation should be separated from the system models. The main research gap should be explained briefly in the problem formulation section with a motivational example/application from real world scenario.

·       The figure 1 looks too generic. Its quality can be enhanced both content wise and visually. Moreover, the figure should be moved inside the text.

·       In result “Ω” shows what? I think you mean system overhead. Mention it in the y-axis of figures.

·       It is not clear that on what network performance parameters the Ω or system overhead depends. Clearly mention in the results section. It is recommended to use the standard parameters that are used in the MEC paradigm instead of generic ones like “system overhead”.

·       What are the limitations of your model? I could not find it anywhere. For example, nature inspired algorithms like GA, PSO and SA have higher search space and complexity. How you addressed the complexity of your approach that is PSO based? Pls add in the manuscript.

·       The future research directions are missing. Pls discuss them in conclusion.

·       Since the authors are not native English speakers. It is suggested that 1-2 native English experts should proofread the article for corrections.

·       Some of the references are old. Pls update them and include the highly related works on task offloading in MEC. For example a few are given below that can be cited positively.

LiMPO: lightweight mobility prediction and offloading framework using machine learning for mobile edge computing.

COME-UP: Computation Offloading in Mobile Edge Computing with LSTM Based User Direction Prediction.

Round 2

Reviewer 3 Report

The previous concerns are well addressed. The paper may be accepted for publication